# Statistical Depth for Ranking and Characterizing Transformer-Based Text Embeddings

**Parker Seegmiller** and **Sarah Masud Preum**
Department of Computer Science
Dartmouth College
Hanover, NH, USA
{pkseeg.gr, Sarah.Masud.Preum}@dartmouth.edu

## Abstract

The popularity of transformer-based text embeddings calls for better statistical tools for measuring distributions of such embeddings. One such tool would be a method for ranking texts within a corpus by centrality, i.e. assigning each text a number signifying how representative that text is of the corpus as a whole. However, an intrinsic center-outward ordering of high-dimensional text representations is not trivial. A *statistical depth* is a function for ranking $k$-dimensional objects by measuring centrality with respect to some observed $k$-dimensional distribution. We adopt a statistical depth to measure distributions of transformer-based text embeddings, *transformer-based text embedding (TTE) depth*, and introduce the practical use of this depth for both modeling and distributional inference in NLP pipelines. We first define TTE depth and an associated rank sum test for determining whether two corpora differ significantly in embedding space. We then use TTE depth for the task of in-context learning prompt selection, showing that this approach reliably improves performance over statistical baseline approaches across six text classification tasks. Finally, we use TTE depth and the associated rank sum test to characterize the distributions of synthesized and human-generated corpora, showing that five recent synthetic data augmentation processes cause a measurable distributional shift away from associated human-generated text.

## 1 Introduction

The transformer architecture (Vaswani et al., 2017) has revolutionized the field of natural language processing (NLP). Generalized transformer-based text embedding models such as S-BERT (Reimers and Gurevych, 2019) and GenSE+ (Chen et al., 2022) have yielded state of the art performance results on a variety of tasks such as natural language inference (NLI) (Williams et al., 2018) and semantic textual similarity (STS) (Cer et al., 2017). The

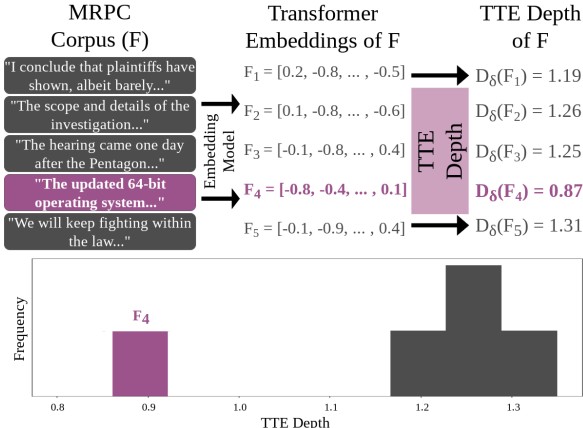

Figure 1: TTE depth gives a center-outward ordering of texts within a corpus using their representations in embedding space. This statistical tool can be used in modeling tasks and inference tasks. Examples from the Microsoft Research Paraphrase Corpus (Dolan and Brockett, 2005) demonstrate that representative samples from the corpus receive high depth scores and outliers receive low depth scores.

goal of these models is to provide a semantically-meaningful, extrinsically-capable vector representation for a given text document of any length. However, these representations do not immediately provide a natural ordering or notion of center with respect to a corpus distribution, which would aid in tasks like outlier detection (see Figure 1). Many transformer-based NLP pipelines would benefit from such a center-outward ordering of texts within a corpus and related statistical inference, for example prompt selection for in-context learning (Gao et al., 2020) or characterizing distributional differences between embeddings of texts from two corpora (Pillutla et al., 2021). An intrinsic notion of centrality (i.e. representation with respect to the corpus distribution) and outlyingness in $k$-dimensional embedding space is not trivial.

A **statistical depth** is a function for ordering multidimensional objects with respect to some observed distribution. Depths have been defined for

functional data (López-Pintado and Romo, 2009), directional data (Pandolfo et al., 2018), and statistical representations of text data (Bolívar et al., 2023). In addition to offering an ordering for a single collection of multidimensional objects, a statistical depth also enables testing for significant differences between two distributions of such objects. We introduce a statistical depth for *T*ransformer-based *Text E*mbeddings, *TTE depth*, which provides a center-outward ordering of texts within a corpus based on the distributional properties of the texts in embedding space. A diagram of this tool is given in Figure 1. While there exist approaches for characterizing corpus distributions such as perplexity (Holtzman et al., 2019) and more recent approaches like MAUVE (Pillutla et al., 2021), TTE depth is an orthogonal statistical tool that gives an intuitive centrality ranking for use in both modeling and distributional inference in NLP pipelines. We investigate the use of depth in both *modeling*, via the task of prompt selection for in-context learning, and *distributional characterization*, via measuring the distributional differences between synthetic data and human-written data in synthetic data augmentation pipelines. The specific contributions of this work are as follows.

- We introduce the use of statistical depth for transformer-based text embeddings by defining an angular distance-based depth for giving a center-outward ordering of transformer-based embeddings of texts within a given corpus. We call this depth *transformer-based text embeddings (TTE) depth*. Alongside TTE depth we introduce the use of a Wilcoxon rank sum test for determining whether two sets of transformer-based embeddings can reasonably be assumed to come from significantly different distributions. This test allows researchers to both characterize distributional shifts between corpora and determine whether such a shift is significant. As the goal of this work is to introduce TTE depth as a useful new tool in both modeling and distributional inference, we release a simple python package[1] to facilitate the use of TTE depth in NLP pipelines.

- To demonstrate the potential of TTE depth in NLP modeling tasks, we use TTE depth as a ranking for the task of prompt selection for in-context learning across various text clas-

sification tasks. Prompt selection is the task of selecting representative examples from a labeled training dataset to prompt a large language model (LLM) for labeling an unseen test sample (Gao et al., 2020). We show that ranking via TTE depth improves F1 performance over statistical baselines in 5 out of 6 text classification tasks.

- As TTE depth is also a statistical inference tool for characterizing corpus distributions, we use it to measure the differences between the embedding distributions of 5 human-written datasets and related synthetically augmented datasets. Measuring the distance between human-written and synthetic corpora is a critical problem in the era of LLMs (Pillutla et al., 2021). We show that TTE depth can reliably determine when human-written and synthetic corpora differ significantly, and thoroughly investigate a case study using the synthetic natural language inference corpus, SyNLI (Chen et al., 2022).

## 2 Related Works

### 2.1 Distributional Characterization of Text Embeddings

A common task in NLP pipelines is to characterize text documents using text metrics, either for error analysis or other research purposes (Hansen et al., 2023). Many statistical linguistics-based metrics exist, such as perplexity (Holtzman et al., 2019) or the Flesch reading ease metric (Hansen et al., 2023), which measures the reading comprehension level of a given document. While most existing metrics are focused on text-level statistics, a few tools exist for comparing transformer-based text embeddings and other deep learning embeddings.

Recent work has defined out-of-distribution samples for image classification (Kaur et al., 2023). Additionally, distributional shifts of machine learning training data have been studied using classifier two-sample tests (Jang et al., 2022). While our work similarly measures distributions of deep learning model inputs, we focus on a statistical approach for transformer-based text embeddings.

A recent tool similar to TTE depth is MAUVE (Pillutla et al., 2021), an approach to measuring distributions of LLM-generated texts using divergence frontiers. While similar in goal, MAUVE and TTE depth differ in that TTE depth is a tool

---

[1]https://github.com/pkseeg/tte_depth

for ordering text embeddings rather than measuring token distributions of text itself.

Coincidentally, MAUVE is built using Kullback-Leibler (KL) divergence (Joyce, 2011), a statistical method for pairwise comparison of distributions, suitable for text embeddings. While several pairwise comparison tools such as KL divergence exist for comparing text embeddings, as far as we are aware TTE depth is the first tool of its kind for providing corpus-level measures of centrality and spread in transformer-based text embedding space.

## 2.2 Depths for Multidimensional Data Objects

Statistical depth is a tool for characterizing sets of multidimensional objects. Depth measures assign each object in a set a measure of centrality with respect to the observed distribution of the set, thus an interpretable center-outward ordering. Depth has been applied to measure a variety of multi-dimensional data (Dai et al., 2022). As angular measures of text embeddings are often used to describe how texts are related, depths for directional data are particularly relevant to our work. Recently depths have been defined for and applied in directional data (Pandolfo et al., 2018; Pandolfo and D'Ambrosio, 2021). While depth has shown to be an effective, interpretable method of endowing multidimensional distributions with notions of centrality and outlyingness, we are the first to use statistical depth for transformer-based embeddings of text data.

Perhaps most similar to our goal is a recent work which defines compositional depth, a depth for text corpora in which an inverse Fourier transform is applied to the *tf-idf* embeddings of texts within a corpora and a statistical depth for functional data is used for ranking (Bolívar et al., 2023). The authors use this depth for classification of health texts, showing comparable performance to other statistical text classification methods. While our work is similar in that we are applying a notion of depth to text embeddings, (as far as we are aware) we are the first to apply statistical depth to transformer-based text embeddings and to highlight use cases of such a depth measure in modern NLP pipelines.

## 3 Transformer-based Text Embedding Depth

Angular distance properties of text embeddings, such as high cosine similarity between embeddings of two similar texts, are often sought as a desired property of such embeddings (Reimers and Gurevych, 2019). As such, we utilize an angular distance-based depth for our investigation. A class of angular distance-based depths for directional data was first introduced by (Pandolfo et al., 2018), and later used by (Pandolfo and D'Ambrosio, 2021) for classification of directional data. Leaning on two depth definitions as introduced there we define a depth for transformer-based text embeddings, TTE depth.

**TTE depth:** Consider a set of embedded texts $\mathcal{F}$ derived from the theoretical set of all possible such documents $S$, each embedded by some text embedding model $M : S \to \mathbb{R}^k$ which embeds texts into vectors of length $k$. Given a bounded distance $\delta(\cdot, \cdot) : \mathbb{R}^k \times \mathbb{R}^k \to \mathbb{R}$, the TTE depth of an embedded text $x \in \mathcal{F}$ with respect to $\mathcal{F}$ is defined as

$$D_\delta(x, \mathcal{F}) := 2 - E_\mathcal{F}[\delta(x, H)] \qquad (1)$$

where $H \sim \mathcal{F}$ is a random variable with uniform distribution over $\mathcal{F}$. Note that while $M$ could be *any* text embedding model, in this work we specifically investigate transformer-based embedding methods. TTE depth can be used in conjunction with two bounded distance functions.

The **cosine distance** $\delta_{cos}(\cdot, \cdot)$ between two embedded text documents $x, y \in \mathcal{F}$ is well explored as a tool for examining pairwise relationships between transformer-based text embeddings (Sunilkumar and Shaji, 2019). It is defined as

$$\delta_{cos}(x, y) := 1 - \frac{x \cdot y}{\|x\| \|y\|} \qquad (2)$$

For any two embedded texts $x, y \in \mathcal{F}$, the **chord distance** $\delta_{ch}(\cdot, \cdot)$ between the two is defined as

$$\delta_{ch}(x, y) := \sqrt{2(1 - x'y)} \qquad (3)$$

While we primarily investigate TTE depth using cosine distance, either cosine distance or chord distance can be used for calculating TTE depth.

Using either distance function, we define the **median** of $\mathcal{F}$, denoted $x_0$, to be the text embedding in $\mathcal{F}$ with maximum depth. In other words, $\max_{x \in \mathcal{F}} D_\delta(x, \mathcal{F}) = D_\delta(x_0, \mathcal{F})$. TTE depth produces this well-defined median at the center of the set of text embeddings $\mathcal{F}$. Ordering the text embeddings in $\mathcal{F}$ by TTE depth gives a center-outward ranking of all text embeddings within a corpus.

**Wilcoxon Rank Sum Test:** As TTE depth gives an ordering to a corpus of text embeddings, we

can use an existing rank sum test for determining whether the embeddings of two corpora differ significantly. Existing applications of depth utilize the Wilcoxon rank sum test generalized to multivariate data through the order induced by a depth function (López-Pintado and Romo, 2009; Liu and Singh, 1993); we follow these works in our definitions.

Suppose we have two corpora, and the sets of the embeddings in $\mathbb{R}^k$ of texts within these corpora are given by $F$ and $G$. We wish to determine whether it is unlikely that $G$ comes from the same distribution as $F$. For a given text embedding $y \in G$, we first define $R(y, F)$ as the fraction of the $F$ population which is "less central" to $F$ than the embedding $y$. That is, let $X \sim F$ be a random variable with uniform distribution over $F$ and define

$$R(y, F) = Pr[D_\delta(X, F) \leq D_\delta(y, F)] \quad (4)$$

This roughly measures how central a given text embedding $y \in G$ is with respect to text embeddings in $F$. For example, if $R(y, F)$ is low, then a large portion of the text embeddings in $X$ are more central to $F$ than $y$, indicating that $y$ is an outlier with respect to $F$. Next we use $R(y, F)$ to define the $Q$ parameter to gauge the overall "outlyingness" of the $G$ text embeddings with respect to $F$. Let $X \sim F$ and $Y \sim G$ be independent random variables with uniform distributions over $F$ and $G$, and define

$$Q(F, G) = Pr[D_\delta(X, F) \leq D_\delta(Y, F)] \quad (5)$$

Since $R(y, F)$ is the fraction of the $F$ corpus which is "less central" to $F$ than the value $y$, $Q(F, G)$ is the average of such fractions over all $y$'s from the $G$ corpus. This means that $Q$ allows us to characterize and quantify how two corpora differ in embedding space. When $Q < \frac{1}{2}$, it means on average more than 50% of the $F$ corpus is deeper than text embeddings $Y$ from $G$, indicating $Y$ is more outlying than $X$ with respect to $F$, hence an inconsistency between $F$ and $G$.

We use the $Q$ parameter to test whether two sets of embedded texts $F$ and $G$ are likely to come from the same distribution. (Liu and Singh, 1993) propose a Wilcoxon rank sum test for the hypotheses $H_O : F = G$ versus $H_a : Q < \frac{1}{2}$. Let $X = \{x_1, x_2, ..., x_m\}$ and $Y = \{y_1, y_2, ..., y_n\}$ be samples of text embeddings from $F$ and $G$, respectively. A sample estimate of $Q$ is $Q(F_m, G_n) =$

$\frac{1}{n} \sum_{i=1}^{n} R(y_i, F_m)$, where $R(y_i, F_m)$ is the proportion of $x_j$'s having $D_\delta(x_j, F_m) \leq D_\delta(y_i, F_m)$. In other words, $Q(F_m, G_n)$ is the average of the ranks of $y_i$'s in the combined sample $X \cup Y$. If this sample estimate is low, we consider the likelihood that the samples $X$ and $Y$ come from the same distribution of text embeddings to be low, and say that it is instead likely that $F$ and $G$ differ significantly. As empirically shown in (Liu and Singh, 1993), the null distribution of $W = [Q(F_m, G_n) - Q(F, G)]$ is approximated by $\mathcal{N}(0, \frac{\frac{1}{m} + \frac{1}{n}}{12})$, and a $Z$ test can be used from there. We give a detailed example of how to interpret the $Q$ parameter and the Wilcoxon rank sum test in Section 5.2.

## 4 Methods

To investigate the usefulness of TTE depth as both a modeling and distributional inference tool, we perform several experiments. We first develop an intuition as to how many texts should be sampled from a population to ensure quality TTE depth results, and use this intuition throughout our evaluation. Next, we explore the use of TTE depth as a ranking technique for in-context learning prompt selection across six classification tasks. Finally, we use TTE depth and the associated rank sum test to examine the distributional differences between five {human-written dataset, synthesized dataset} pairs.

### 4.1 Sample Size Recommendations

We wish to develop an intuitive understanding of how many samples are required for consistent TTE depth measures, estimates of the $Q$ parameter, and Wilcoxon rank sum test results. While it is theoretically possible to obtain these metrics with access to full corpora, we choose to investigate sampling techniques for two reasons. First, as TTE depth requires a pairwise distance comparison across a corpus, it becomes computationally intractable to rank each text within a large corpus. Second, there exist situations in which obtaining exorbitantly large samples of two corpora for comparison is expensive or time-consuming, for example, engineers who wish to measure a distributional shift in responses of a deployed LLM. Hence, we are interested in how many samples $n$ are required from two corpora $F$ and $G$ in order to get a good estimate $Q(F_n, G_n)$ for $Q(F, G)$, thus a good TTE depth measurement.

We run a simulation study on the Wilcoxon rank sum test for corpora on the order induced by TTE

depth. We use the natural language inference (NLI) corpus (Bowman et al., 2015) and an associated synthetic NLI (SyNLI) corpus (Chen et al., 2022) for this study. We sample 5,000 texts from each of the NLI and SyNLI corpora pair and treat them as the population corpora $F$ and $G$, respectively. We then embed each text in each population corpus using S-BERT, a popular general-purpose text embedding model (Reimers and Gurevych, 2019)[2]. We then use the order induced by TTE depth to compute the "true" $Q$ parameter of these populations, $Q(F, G)$. To empirically determine a good sample size $n$ to draw from $F$ and $G$ for estimating $Q(F, G)$ with $Q(F_n, G_n)$, we iterate over sample sizes $n \in \{5, 25, 50, 100, 500\}$, randomly draw samples of size $n$ from $F$ and $G$, and compute $Q(F_n, G_n)$ 20 times. We are interested in whether the distribution of $Q(F_n, G_n)$ accurately represents the true value of $Q(F, G)$ consistently with different sample sizes.

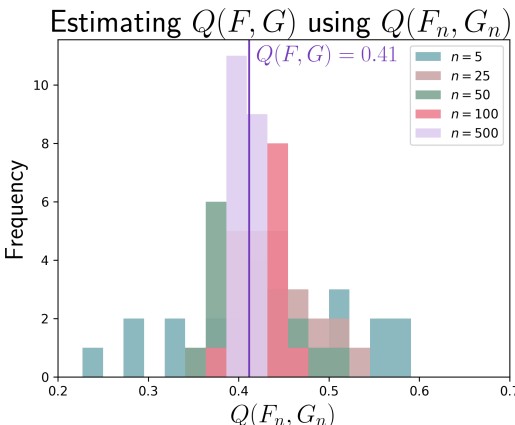

Figure 2: Comparison of $Q(F_n, G_n)$ estimates for different values of $n$. As expected, we see the center of each sample distribution centered around the true $Q(F, G)$. Additionally, we see the spread of the distribution of $Q(F_n, G_n)$ estimates decreases as sample sizes increases. As the distribution of $Q(F_n, G_n)$ with $n = 500$ has low variance, we henceforth use $n = 500$ for estimating $Q(F, G)$ in the Wilcoxon rank sum test.

The true value of $Q(F, G)$ was found to be 0.4118. We can see the sample distributions of $Q(F_n, G_n)$ for different values of $n$ in Figure 2. As expected, each of the distributions of $Q(F_n, G_n)$ estimates is centered around $Q(F, G)$, and the spread of the sample distribution decreases as the

---

[2]While TTE depth can be used in conjunction with any text embedding model and either bounded distance, we use only S-BERT and cosine distance in our evaluations for simplicity. A comparison of different embedding models and bounded distances is provided in Appendix A.1.

sample size increases. We note that the distribution of $Q(F_{500}, G_{500})$ estimates has a standard deviation of just 0.009, indicating low variance around the true value of $Q(F, G) = 0.4118$ while running in a reasonable amount of time. We propose that $n = 500$ is a good enough sample size to test for significantly different distributions of two sufficiently large embedded corpora using the Wilcoxon rank sum test on the order induced by TTE depth, and we sample $n = 500$ texts from each corpus for the evaluations in this work.

## 4.2  In-Context Learning Prompt Selection

In-context learning (ICL) is a recent paradigm used for few-shot NLP tasks. In text classification it involves prompting a pre-trained LLM to assign a label to an unseen input after being given several examples (Gao et al., 2020). Recent work has investigated prompt selection, i.e. the task of selecting representative examples from a labeled training dataset to prompt a LLM for labeling an unseen test sample (Gao et al., 2020). We propose the use of TTE depth for the task of prompt selection in a model-agnostic setting. Exploration into ICL has shown that an important component of ICL prompting is the inclusion of in-distribution texts in the ICL prompt (Min et al., 2022). We hypothesize that texts which better represent the corpus distribution, i.e. texts with higher TTE depth, will improve downstream ICL performance. Since TTE depth is a statistical ranking procedure, we do not compare extrinsic results against existing models for prompt selection. Instead, we extrinsically evaluate test two TTE depth-based rankings against two naive statistical baseline rankings across six text classification tasks.

*Random (RAND)* ranking simply randomly shuffles the training dataset, after which the first $N$ texts are used as ICL examples for prompt selection.

*Label Distribution Match (LDM)* ensures that the label distribution of the training dataset is matched by the selected ICL examples. For each label occupying $l\%$ of the training data label space, after applying a random shuffle the first $N \cdot l\%$ texts with that label are selected as ICL examples. This ranking attempts to ensure that the LLM is able to accurately infer the label distribution of the training data through the ICL examples.

*Depth (DEEP)* ranking ranks all embedded texts by their TTE depth. The $N$ deepest texts are then selected as ICL examples.

*Depth with Label Distribution Match (DLDM)* ranking is a combination of both the DEEP and LDM techniques. This ranking first ranks all embedded texts by their TTE depth. Then, the same selection strategy as the label distribution match ranking is used. Hence, the $N \cdot l\%$ deepest texts for each label are selected as ICL examples.

We compare ranking methods across six text classification tasks. The Corpus of Linguistic Acceptability *(COLA)* (Warstadt et al., 2018), Microsoft Research Paraphrase Corpus *(MRPC)* (Dolan and Brockett, 2005), Stanford Sentiment Treebank *(SST2)* (Socher et al., 2013), and Recognizing Textual Entailment *(RTE)* (Bentivogli et al., 2017) tasks are selected from the GLUE benchmark (Wang et al., 2019). The final 2 tasks, the tweet hate speech detection *(THATE)* (Basile et al., 2019) and tweet offensive language identification *(TOFF)* (Zampieri et al., 2019) tasks are both tweet classification tasks.

For each task, we randomly sample 500 texts from the training corpus to represent the labeled training dataset for prompt selection. We embed each of these texts (for pairwise tasks we embed a joined version of the two texts) using S-BERT. Next, we rank each embedded text according to each of the 4 baseline and TTE depth-based ranking strategies described. The $N$ ICL examples given by each ranking strategy are then used to create the ICL prompt. We use this prompt to query ChatGPT, a popular LLM which has been shown to offer good performance across several ICL text classification tasks (Ray, 2023), to label each evaluation text in the evaluation corpus of each task. After post-processing ChatGPT output, results are recorded for each ranking strategy and are averaged over three values of $N$ for each task.

### 4.3 Distributional Characterization of Synthesized Training Datasets

The theory of data synthesis for training or fine-tuning language models is straight-forward: more data is good for learning, and synthesized data is cheaper and more accessible than human-generated and human-labeled data (He et al., 2022). However, most data synthesis techniques report only extrinsic evaluations of the synthesized data, i.e. an increase in F1 on hidden evaluation data. Intrinsic evaluations, such as the *distributional effects* of synthesizing training samples for augmenting NLP datasets, require further exploration (Pillutla et al.,

2021). Specifically, we're interested in whether synthesized training texts across various NLP tasks are significantly different from their human-written training text counterparts in embedding space. We use TTE depth to investigate the following synthesis strategies.

*Inversion, Passivization, and Combination* are NLI sample synthesis techniques involving swapping the subjects and objects of sentences in the origin dataset, taking active verbs in sentences and converting them to passive verbs, and both simultaneously (Min et al., 2020). These synthesized samples are obtained by inverting samples from the Multi NLI (MNLI) corpus from the GLUE benchmark (Williams et al., 2018), hence we use TTE depth to measure the distributional shift of the inverted corpus against MNLI in embedding space. To embed sentence pair samples in the NLI task, we join them and embed them afterwards.

*SyNLI* is a dataset of synthetic NLI samples generated using a generator/discriminator model trained on the NLI corpus (MNLI + Stanford NLI corpus (Bowman et al., 2015)) to synthesize sentence pairs (Chen et al., 2022). SyNLI sentence pairs are generated using T5, a popular open-source LLM (Raffel et al., 2020), and are compared against samples from NLI in our TTE depth analyses.

*SRQA-AUTO* is a dataset of stories involving spatial relationships between objects and spatial reasoning questions corresponding to the stories (Mirzaee et al., 2021). In SRQA-AUTO the stories are generated from image captions using context-free grammars (CFGs) and a rule base is used to generate the corresponding spatial reasoning questions. In our evaluation, SRQA-AUTO is compared against the SRQA-HUMAN dataset containing human-written {story, question} pairs. Samples in SRQA-AUTO and SRQA-HUMAN are embedded by joining story and question pairs and embedding the resulting text, and these embeddings are used in our TTE depth analysis.

For each training data synthesis technique, we use TTE depth to rank a sample of 500 synthesized texts and 500 texts from the corresponding human-written corpus. We compare the depth-induced distributions intuitively, reporting the TTE depth of the median of the human-written text (*Med Human*) and the synthesized text (*Med Synth*) with respect to the human-written embedding distribution. We then use the Wilcoxon rank sum test on the order

induced by TTE depth to compare the distributions statistically, testing whether the human-written and synthesized corpora differ significantly in embedding space and giving $Q$ estimates, $W$ statistics, and $p$ values. We further detail one analysis, examining the $Q$ measure of outlyingness obtained through TTE depth and discussing what it means for the synthesis technique to produce more outlying texts than the human-written corpus.

# 5 Results and Discussion

## 5.1 TTE Depth for In-Context Learning Prompt Selection

Prompt selection for ChatGPT is a task with a black box evaluation setting due to opaque training processes, hence ICL results can be somewhat random (Ji et al., 2023). However, we hypothesize that since TTE depth will lead to prompts which are more representative of the labeled training dataset, on average these prompts should lead to better in-context tuning for LLMs such as ChatGPT.

Results of the prompt selection evaluation described in Section 4.2 are displayed in Table 1. We report average F1 scores across $N = \{5, 8, 11\}$ samples selected using each ranking strategy. We use McNemar's test to determine whether depth-based strategies significantly outperform baselines (McNemar, 1947). For each of the depth-based strategies, we report whether the results were statistically significantly better than each of the baseline strategies (p < 0.05) using the notation (Y/N) for whether the results were significantly better than the RAND and LDM baselines, respectively. Across the six text classification tasks, TTE depth-based ranking strategies outperform the naive baselines in five tasks and the DLDM ranking outperforms all other ranking strategies on average. Additionally, the DLDM ranking strategy outperforms both baselines on average. In 9 of the 24 pairwise comparisons (including all 8 across the 2 social media tasks) the performance of depth-based ranking strategies is statistically significantly better than baseline approaches.

## 5.2 Distributional Characterization of Synthesized NLP Training Datasets

Results of our analysis are found in Table 2. As described in Section 4.3, we give the median depth values of each human-generated and synthetic corpus, along with $Q$ parameter estimates, the Wilcoxon rank sum test statistic $W$, and the

associated $p$-value. According to the Wilcoxon rank sum test on the order induced by TTE depth, all of the training sample synthesis techniques we evaluate produce a significantly different distribution of texts in embedding space. This indicates that some aspect of each synthesis process causes the synthesized training samples to be either generally more central or more outlying than the origin training samples. Within the NLI task, the SyNLI synthetic dataset results in the lowest $Q$ estimate, indicating that it is the furthest in embedding space from its origin dataset. This result is intuitive–the other three NLI synthesis strategies simply result in transposed versions of the original text, whereas the SyNLI text samples are entirely new samples generated by an LLM. We briefly focus on this test as a thorough example of using TTE depth as a corpus analysis tool.

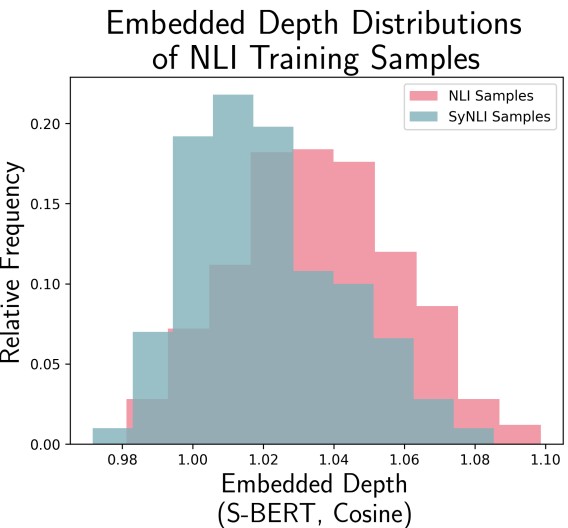

Figure 3: Comparison of TTE depth distributions of samples from the NLI and SyNLI datasets, with respect to the NLI embedding distributions. The NLI and SyNLI corpora are differ significantly according to the Wilcoxon rank sum test, hence the synthesis process used to generate samples in the SyNLI dataset significantly shifts the distribution away from the NLI corpus.

Examining row 4 of Table 2 we find that the TTE depth ranking of SyNLI texts with respect to the distribution of NLI texts results in a $Q$ estimate of 0.4064. A direct interpretation of this parameter means that, on average, texts from the SyNLI corpus are more central (i.e. closer to the TTE depth median of the NLI corpus) than only 40.64% of the the texts in the NLI corpus. As described in Section 3, a $Q$ parameter this low indicates that the texts in the SyNLI corpus mark a clear distributional shift

| Task | Random | Label Distribution Match | Depth-First | Depth-First Label Distribution Match |
|---|---|---|---|---|
| COLA | 0.8164 | 0.8094 | 0.8166 (N/N) | **0.8220 (N/Y)** |
| MRPC | 0.7166 | 0.7250 | **0.7334** (N/N) | 0.6841 (N/N) |
| SST2 | 0.9486 | 0.9514 | **0.9518** (N/N) | 0.9414 (N/N) |
| RTE | 0.6366 | **0.7352** | 0.4413 (N/N) | 0.7258 (N/N) |
| THATE | 0.6039 | 0.6629 | **0.7072 (Y/Y)** | 0.7009 **(Y/Y)** |
| TOFF | 0.7361 | 0.7189 | **0.7461 (Y/Y)** | 0.7411 **(Y/Y)** |
| AVG | 0.7431 | 0.7671 | 0.7332 | **0.7692** |

Table 1: Results of the TTE depth-based naive ranking strategies across the six text classification tasks. The TTE depth-based ranking strategies outperform the naive baselines in 5 of the 6 tasks, and the DLDM ranking outperforms all other ranking strategies on average.

| Human Corpus | Synth Corpus | Med Human | Med Synth | $Q$ | $W$ | $p$ |
|---|---|---|---|---|---|---|
| MNLI | Inversion | 1.0431 | 1.0397 | 0.4751 | 2.67 | 0.0077 |
| MNLI | Passivization | 1.0431 | 1.0408 | 0.4771 | 2.45 | 0.0142 |
| MNLI | Combination | 1.0431 | 1.0397 | 0.4785 | 2.30 | 0.0214 |
| *MNLI + SNLI* | *SyNLI* | *1.0339* | *1.0175* | *0.4064* | *10.19* | *< 0.0001* |
| SRQA-HUMAN | SRQA-AUTO | 1.8459 | 1.7756 | 0.3131 | 20.36 | < 0.0001 |

Table 2: Embedded depth and Wilcoxon rank sum test results comparing human-written and synthesized data across five data synthesis techniques. For each test, a sample of 500 texts was taken from each of the origin and synthesized training datasets. Each text was embedded using S-BERT, and a Wilcoxon rank sum test on the order induced by embedded depth was performed. For each test, the $Q(F_{500}, G_{500})$ estimate is given, along with the corresponding $W$ statistic and associated $p$-value. A low $p$-value indicates that it is unlikely the origin and synthesized datasets are drawn from the same distribution in embedding space.

away from the NLI corpus.

We can see this distribution shift graphically in Figure 3, where the distributions of TTE depths for the randomly sampled SyNLI texts and NLI texts, with respect to the NLI texts, are separated. This indicates that the synthesis technique used to create the SyNLI corpus produces texts which are distant from the texts in the human-written NLI corpus, as measured by cosine similarity between S-BERT embeddings. This distributional shift can be seen in the $Q$ parameter as well as the centers of the TTE depth distributions: the median depth of the NLI texts (1.0339) is higher than the median depth of the SyNLI texts (1.0175). This shift could cause unintended consequences if the SyNLI corpus were to be treated equally with the human-written NLI corpus in downstream applications.

TTE depth and the associated Wilcoxon rank sum test have the potential to be useful in both modeling and distributional inference tasks. TTE depth allows NLP researchers and practitioners to measure distributional properties of corpora such as center and spread, as well as compare distributions of corpora in embedding space.

## 6 Limitations and Future Work

As defined in Section 3, TTE depth depends on text embedding models which produce embeddings with desired spatial relationships. This assumption holds for most popular text embedding models, including those used in this work. However, text vectorizations which do not claim this property, such as *tf-idf*, are not suitable for use in TTE depth. Additionally, TTE depth at its core is a dimensionality reduction method that produces a single interpretable dimension along which texts can be measured. We note that no matter how the dimensionality of text embeddings is reduced, there will always be some amount of information loss. While the single dimension and intuitive center-outward ordering given by TTE depth is useful for many tasks, it may not be appropriate as a one-size-fits-all approach to distributional inference and is meant to be used in conjunction with other statistical tools for a full picture of embedding space properties.

Future work may use TTE depth for a variety of NLP modeling and inference tasks. Modeling tasks may include the use of TTE depth to select representative documents to facilitate hierarchical multi-

document summarization (Fabbri et al., 2019). Inference tasks could include the use of TTE depth to investigate outlying documents in domain-specific corpora, such as research in anomaly detection in social media posts (Guha and Samanta, 2021).

# 7 Conclusion

In this work we have introduced TTE depth, a novel approach to measure distributions of transformer-based text embeddings which endows corpora with intuitive notions of center and spread. TTE depth gives a center-outward ordering of texts within a corpus based on the texts in embedding space. We have defined TTE depth and an associated Wilcoxon rank sum test for discovering significant differences between pairs of corpora. To demonstrate the use of TTE depth in modeling, we have used TTE depth as a ranking for the task of prompt selection, showing good performance across a variety of text classification tasks. Additionally, we have shown the usefulness of TTE depth for characterizing distributional properties of corpora by investigating the embedding distributions of source and synthetic datasets in several synthetic data augmentation pipelines. TTE depth is a useful, novel tool for both modeling and distributional inference in modern transformer-based NLP pipelines.

# Acknowledgements

This work was supported by the National Science Foundation Research Traineeship, Transformative Research and Graduate Education in Sensor Science, Technology and Innovation (DGE- 2125733).

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

# A Appendix

## A.1 Comparison of Embedding Models and Bounded Distances

While in our evaluations, we investigate TTE depth using only S-BERT, we also provide an investigation of TTE depth under different embedding models here. We also further investigate TTE depth using chord distance, in addition to cosine distance. To test whether $Q$ estimates vary across embedding models and distance functions, we use the SyNLI and NLI corpora pair as in 4.1. For each embedding model and distance function, we sample 100 texts from each of the SyNLI and NLI corpora, compute embeddings using the embedding model, and compute TTE depth using the distance function 20 times. As in our sample size analysis, we are interested in whether the distribution of $Q(F_{100}, G_{100})$ estimates varies depending on the embedding method and distance function used. In addition to the 2 bounded distances (cosine and chord distance), we compare 4 popular embedding models.

*Sentence BERT (S-BERT)* is a popular general-purpose text embedding model which is trained using a triplet network structure (Reimers and Gurevych, 2019). To get a sense of whether the size of the embedding model matters for TTE depth results, we use both S-BERT base and S-BERT large.

*SimCSE* is a text embedding model trained on labeled NLI datasets with a contrastive learning objective to learn a general text embedding model (Gao et al., 2021). At the time of publication, SimCSE achieved SOTA results on semantic textual similarity (STS) benchmarks. We utilize both the BERT-based and RoBERTa-based SimCSE models in our analysis.

*GenSE+* is a text embedding method trained in part using the SyNLI dataset we use for TTE depth analysis (Chen et al., 2022). Upon release, GenSE+ achieved SOTA performance on STS benchmarks, making it a good embedding model for use in TTE depth analysis.

*MPNET* uses a novel masked and permuted pre-training method for developing a text embedding model (Song et al., 2020). MPNET reports SOTA performance on a variety of downstream tasks.

Table 3 shows the mean and standard deviation of 20 $Q(F_{100}, G_{100})$ estimates on the order induced by TTE depth across different bounded distance functions and embedding models, where $F$ is the NLI corpus and $G$ is the SyNLI corpus.

As discussed in Section 3, a lower $Q$ indicates that the embeddings in corpus $G$ are more outlying on average than embeddings in corpus $F$, with respect to the center of $F$. Intuitively, the embedding models with the best reported performance on STS benchmarks (MPNET and GenSE+) have the lowest average $Q$ estimates, indicating that they are better able to discriminate between NLI and SyNLI in embedding space. Additionally, within the embedding models the $Q$ estimates are very similar across the two distance functions. On average, $Q$ estimates using chord distance are only slightly lower than estimates using cosine distance with an average difference of 0.0041. While MPNET and GenSE+ produce more desirable $Q$ estimates on average, each transformer-based embedding model produces roughly the same $Q$ estimate with $n = 100$, indicating that any combination is suitable for TTE depth analysis.

| Distance Function | Embedding Model | Embedding Dimension | Mean $Q$ | Standard Deviation $Q$ |
|---|---|---|---|---|
| Cosine | S-BERT Base | 384 | 0.4158 | 0.0233 |
| Chord | S-BERT Base | 384 | 0.4118 | 0.0233 |
| Cosine | S-BERT Large | 384 | 0.4219 | 0.0283 |
| Chord | S-BERT Large | 384 | 0.4176 | 0.0278 |
| Cosine | MPNET | 768 | 0.3905 | 0.0242 |
| Chord | MPNET | 768 | 0.3878 | 0.0236 |
| Cosine | SimCSE BERT | 768 | 0.4309 | 0.0233 |
| Chord | SimCSE BERT | 768 | 0.4270 | 0.0232 |
| Cosine | SimCSE RoBERTa | 768 | 0.4180 | 0.0186 |
| Chord | SimCSE RoBERTa | 768 | 0.4136 | 0.0185 |
| Cosine | GenSE+ | 768 | 0.4094 | 0.0381 |
| Chord | GenSE+ | 768 | 0.4042 | 0.0370 |

Table 3: Comparison of the mean and standard deviation of 20 $Q(F_{100}, G_{100})$ estimates across different bounded distance functions and embedding models, where $F$ is the NLI corpus and $G$ is the SyNLI corpus.