# OpenReview forum: "Statistical Depth for Ranking and Characterizing Transformer-Based Text Embeddings"
_EMNLP/2023/Conference — EMNLP 2023 Main_

### Official Review · Reviewer_cygC · 2023-08-05

**Soundness:** 3

**Excitement:**

3: Ambivalent: It has merits (e.g., it reports state-of-the-art results, the idea is nice), but there are key weaknesses (e.g., it describes incremental work), and it can significantly benefit from another round of revision. However, I won't object to accepting it if my co-reviewers champion it.

**Paper Topic And Main Contributions:**

This paper introduces a novel method for computing distributions of a corpus using transformer embeddings. It extends the concept of statistical depth, which originally maps objects into a distributional space based on surface form information, into the embedding space. The authors employ distance functions, like cosine similarity, to implement their approach. Through their experiments, they demonstrate that sampling with their proposed method outperforms baseline samplings.

The main contributions of this paper is highlighting the importance of statistical depth and its expansion to the embedding space, and its implementation.

**Questions For The Authors:**

1. what Z means in the Eq (1)? is it normal distribution?
2. How can the proposed approach prevent noisy data (junk data) being selected from the corpus?

**Reasons To Accept:**

- Motivation

This paper addresses the importance of understanding corpus level distributions and their correlations on the downstream tasks.

**Reasons To Reject:**

- Evaluation

The claimed performance gain in Table 1 is relatively small over the baselines. Additionally, I have concerns regarding the proposed approach's applicability, as it seems to be suitable mainly for well-constructed benchmark datasets with minimal noisy (junk) data. In real-world datasets, which typically contain a significant amount of noisy data, there is a high probability that the proposed approach may inadvertently select and prioritize such noisy instances.


- Presentation

The organization of the paper could be improved. Section 4, which is mainly about the experiments, could be merged into Section 5. Furthermore, Section 3 needs refinement to provide a clearer understanding for readers. Some notations, such as "Z" in Eq (1), lack their definitions, which might cause confusion and hinder comprehension.

**Reproducibility:**

4: Could mostly reproduce the results, but there may be some variation because of sample variance or minor variations in their interpretation of the protocol or method.

**Reviewer Confidence:**

3: Pretty sure, but there's a chance I missed something. Although I have a good feel for this area in general, I did not carefully check the paper's details, e.g., the math, experimental design, or novelty.

---

> ### Author Rebuttal · Authors · 2023-08-28
>
> # Summary
> We thank the reviewer for their thoughtful review and perspective. We are encouraged that the reviewer recognizes the importance of understanding corpus-level distributions in transformer-based embedding space. We appreciate the reviewer’s feedback on the applicability of TTE depth, our evaluation of TTE depth for the ICL prompt creation task, and our paper’s presentation. We answer specific questions and feedback below.
>
> # Questions
> > **Q1:** Additionally, I have concerns regarding the proposed approach's applicability, as it seems to be suitable mainly for well-constructed benchmark datasets with minimal noisy (junk) data. In real-world datasets, which typically contain a significant amount of noisy data, there is a high probability that the proposed approach may inadvertently select and prioritize such noisy instances.
>
> > How can the proposed approach prevent noisy data (junk data) being selected from the corpus?
>
> **A1:** We appreciate your perspective on this matter. However, we might have different interpretations. We want to share additional details that might shed light on our point of view.
>
> TTE depth is a method designed for the task of filtering noisy texts and finding representative samples from large corpora. By the definition of the angular distance-based TTE depth function described in our work, texts with the highest TTE depth values are the most representative of their corpus in embedding space and those which have the lowest TTE depth values are the least representative. Unless the majority of texts within a given corpus are “noisy,” in which case the corpus is likely unsuitable for many NLP tasks, TTE depth will not highly rank noisy texts within a corpus. On the contrary, TTE depth is designed to separate noisy texts from the majority text distribution.
>
> We also argue that our evaluation of TTE depth as a ranking strategy for ICL prompt creation highlights the applicability of TTE depth in noisy corpora. Social media corpora, such as the offensive tweet detection (TOFF) and hate speech tweet detection (THATE) tasks we used for evaluation, contain more noisy texts as shown in [1]. TTE depth as a ranking strategy leads to statistically-significant, increased performance on downstream ICL classification tasks on both of these corpora.
>
> In addition to the two social media datasets used in our ICL prompt creation evaluation, we argue that the COLA benchmark dataset is suitable for evaluating the applicability of the TTE depth ranking strategy on noisy data. The Corpus Of Linguistic Acceptability (COLA) dataset presents a binary classification task, to determine whether a given sentence is linguistically acceptable (1) or not (0). [2] 30% of sentences in this corpus are labeled as not linguistically acceptable, meaning the dataset contains a significant amount of noisy data. The TTE depth ranking strategy is able to produce representative texts from this corpus which, when used for ICL classification, produce better results than baselines. We believe that our modeling evaluation using this dataset, in addition to the social media datasets, highlights the applicability of TTE depth on noisy data.
>
> Additionally, our results show that TTE depth, in conjunction with the Wilcoxon rank sum test, is able to effectively distinguish human-written texts from synthetic texts. While not a direct application of finding noisy texts in a corpus, we argue that it is highly correlated. In theory and in evaluation, our work shows that TTE depth is effective in the task of filtering noisy texts from large corpora. We leave evaluation of determining the _sensitivity_ of TTE depth to noisy data to future work given the page limit.
>
> > **Q2:** The claimed performance gain in Table 1 is relatively small over the baselines.
>
> **A2:** We appreciate the reviewer’s feedback on the evaluation of the ICL prompt creation use case of TTE depth. To address this feedback, during the rebuttal period we have obtained new results using the full evaluation datasets, as opposed to samples of each evaluation dataset, for each of the 6 text classification tasks described in Section 4.2 of our paper. Additionally, we have statistically analyzed these new results using McNemar’s test [3]. As a reminder for the reviewer, we compare two TTE depth-based ranking strategies, depth-first (DEEP) and depth-first with label distribution matching (DLDM) against two baselines, random (RAND) and label distribution matching (LDM). We report average F1 scores on ICL classification using 5, 8, and 11 samples selected using each ranking strategy. For each of the depth-based strategies, we report whether the results were statistically significantly better than each of the baseline strategies (p < 0.05) using the signifier (Y/N) for whether the results were significantly better than the RAND and LDM baselines, respectively. Results are presented in the table below.
>
> |  Task |   RAND   |     LDM    |             DEEP            |             DLDM            |
> |:-----:|:--------:|:----------:|:---------------------------:|:---------------------------:|
> |       |    F1    |     F1     | F1 (Sig vs RAND/Sig vs LDM) | F1 (Sig vs RAND/Sig vs LDM) |
> | COLA  |  0.8164  |   0.8094   |         0.8166 (N/N)        |     **0.8220** (N/**Y**)    |
> | MRPC  |  0.7166  |   0.7250   |       **0.7334** (N/N)      |         0.6841 (N/N)        |
> | SST2  |  0.9486  |   0.9514   |       **0.9518** (N/N)      |         0.9414 (N/N)        |
> | RTE   |  0.6366  | **0.7352** |         0.4413 (N/N)        |         0.7258 (N/N)        |
> | THATE |  0.6039  |   0.6629   |   **0.7072** (**Y**/**Y**)  |     0.7009 (**Y**/**Y**)    |
> | TOFF  |  0.7361  |   0.7189   |   **0.7461** (**Y**/**Y**)  |     0.7411 (**Y**/**Y**)    |
> | _AVG_ | _0.7431_ |  _0.7671_  |           _0.7332_          |         **_0.7692_**        |
>
> We highlight that TTE depth-based ranking strategies select better samples for downstream ICL classification performance on 5 of the 6 full evaluation datasets. Additionally, the DLDM ranking strategy outperforms both baselines on average. Finally, in 9 of the 24 pairwise comparisons (including all 8 across the 2 noisy social media tasks) the performance of depth-based ranking strategies is statistically significantly better than baseline approaches. We hope that these newly-obtained results using full evaluation datasets, in addition to the statistical significance testing, demonstrate the usefulness of TTE depth in the ICL prompt creation task. We will add these new results to Section 5 of the paper.
>
> We believe that this evaluation, in addition to the theoretical introduction to TTE depth and the evaluation of TTE depth as an inference method described in our paper, are sound contributions to corpus-level distributional inference and modeling in NLP.
>
> > **Q3:** The organization of the paper could be improved. Section 4, which is mainly about the experiments, could be merged into Section 5.
>
> **A3:** Thanks for the insightful comments! We will definitely reconsider the organization of the draft.
>
> > **Q4:** Furthermore, Section 3 needs refinement to provide a clearer understanding for readers. Some notations, such as "Z" in Eq (1), lack their definitions, which might cause confusion and hinder comprehension.
>
> > what Z means in the Eq (1)? is it normal distribution?
>
> **A4:** We agree that Z was probably not the best choice to represent a random variable in a statistics-related paper. As described after the definition of Eq (1), “EF is the expectation under the assumption that Z has distribution F,” meaning that Z simply represents the realized values of the distribution of F, i.e. the term EF[δ(x, Z)] represents the expected distance from the embedded text x to all other embedded texts in F. Should the paper be accepted, we will change the variable name and add a few points of clarification to Section 3 to provide a clearer understanding for the readers.
>
> # Conclusion/References
> Thank you for your thoughtful review and comments. We hope this response satisfies your concerns, please let us know if you have any other questions!
>
> [1] Baldwin, Timothy, et al. "How noisy social media text, how diffrnt social media sources?." Proceedings of the sixth international joint conference on natural language processing. 2013.
>
> [2] Warstadt, Alex, Amanpreet Singh, and Samuel R. Bowman. "Neural network acceptability judgments." Transactions of the Association for Computational Linguistics 7 (2019): 625-641.
>
> [3] McNemar, Q. 1947. Note on the sampling error of the difference between correlated proportions or percentages. Psychometrika, 12(2): 153–157.

---

### Official Review · Reviewer_vaeR · 2023-08-10

**Soundness:** 4

**Excitement:**

4: Strong: This paper deepens the understanding of some phenomenon or lowers the barriers to an existing research direction.

**Paper Topic And Main Contributions:**

The authors propose transformer-based text embedding (TTE) depth, and illustrate the use of this statistical depth on NLP tasks including in-context learning prompt creation and difference measurement between human-written and machine-generated text. The performance of TTE depth shows small improvement over baselines.

**Reasons To Accept:**

1. The paper is well-written and the research topic is interesting. The proposed method along with its associated python package can be a valuable contribution to the community.
2. The experiments yield positive results on the two evaluation tasks.

**Reasons To Reject:**

1. The evaluation on the prompt creation task is limited. By using the ChatGPT API, a larger evaluation set should be obtained quickly without significant cost.

**Reproducibility:**

4: Could mostly reproduce the results, but there may be some variation because of sample variance or minor variations in their interpretation of the protocol or method.

**Reviewer Confidence:**

3: Pretty sure, but there's a chance I missed something. Although I have a good feel for this area in general, I did not carefully check the paper's details, e.g., the math, experimental design, or novelty.

**Typos Grammar Style And Presentation Improvements:**

Line 340, “computing” looks like an excessive word. Line 395 is missing a comma. Line 528, “the median” should be changed to “the median of.”

---

> ### Author Rebuttal · Authors · 2023-08-28
>
> # Summary
> We thank the reviewer for their thoughtful review. We are encouraged that the reviewer sees the transformer-based text embedding (TTE) depth method and its associated python package as valuable contributions to the community, and finds that our evaluations yield positive results. We appreciate the reviewer’s feedback on the evaluation of TTE depth for the in-context learning (ICL) prompt creation task. We answer specific questions and feedback from the reviewer below.
>
> # Questions
> > **Q1:** The evaluation on the prompt creation task is limited. By using the ChatGPT API, a larger evaluation set should be obtained quickly without significant cost.
>
> **A1:** We appreciate the reviewer’s suggestion to obtain larger evaluation sets for the ICL prompt creation task. To address this concern, during the rebuttal period we have obtained new results using the full evaluation datasets, as opposed to samples of each evaluation dataset, for each of the 6 text classification tasks described in Section 4.2 of our paper. As a reminder for the reviewer, we compare 2 TTE depth-based ranking strategies, depth-first (DEEP) and depth-first with label distribution matching (DLDM), against 2 baselines, random (RAND) and label distribution matching (LDM). We report average F1 scores on ICL classification using 5, 8, and 11 samples selected using each ranking strategy. To address concerns about the statistical significance of our results in our evaluation of TTE depth in the ICL classification use case, during the rebuttal period we have also evaluated these results using McNemar’s test [1]. For each of the depth-based strategies, we report whether the results were statistically significantly better than each of the baseline strategies (p < 0.05) using the signifier (Y/N) for whether the results were significantly better than the RAND and LDM baselines, respectively. Results are presented in the table below.
>
>
> |  Task |   RAND   |     LDM    |             DEEP            |             DLDM            |
> |:-----:|:--------:|:----------:|:---------------------------:|:---------------------------:|
> |       |    F1    |     F1     | F1 (Sig vs RAND/Sig vs LDM) | F1 (Sig vs RAND/Sig vs LDM) |
> | COLA  |  0.8164  |   0.8094   |         0.8166 (N/N)        |     **0.8220** (N/**Y**)    |
> | MRPC  |  0.7166  |   0.7250   |       **0.7334** (N/N)      |         0.6841 (N/N)        |
> | SST2  |  0.9486  |   0.9514   |       **0.9518** (N/N)      |         0.9414 (N/N)        |
> | RTE   |  0.6366  | **0.7352** |         0.4413 (N/N)        |         0.7258 (N/N)        |
> | THATE |  0.6039  |   0.6629   |   **0.7072** (**Y**/**Y**)  |     0.7009 (**Y**/**Y**)    |
> | TOFF  |  0.7361  |   0.7189   |   **0.7461** (**Y**/**Y**)  |     0.7411 (**Y**/**Y**)    |
> | _AVG_ | _0.7431_ |  _0.7671_  |           _0.7332_          |         **_0.7692_**        |
>
> We highlight that TTE depth-based ranking strategies select better samples for downstream ICL classification performance on 5 of the 6 full evaluation datasets. Additionally, the DLDM ranking strategy outperforms both baselines on average. Finally, in 9 of the 24 pairwise comparisons (including all 8 across the social media tasks) the performance of depth-based ranking strategies is statistically significantly better than baseline approaches. We hope that these newly-obtained results using full evaluation datasets demonstrate the usefulness of TTE depth in the ICL prompt creation task. We will add these new results to Section 5 of the paper.
>
> We believe that this evaluation, in addition to the theoretical introduction to TTE depth and the evaluation of TTE depth as an inference method described in our paper, are sound contributions to corpus-level distributional inference and modeling in NLP.
>
> > **Q2:** Line 340, “computing” looks like an excessive word. Line 395 is missing a comma. Line 528, “the median” should be changed to “the median of.”
>
> **A2:** Thank you for your attention to detail. We will fix these typos in the next iteration of our paper.
>
> # References
> [1] McNemar, Q. 1947. Note on the sampling error of the difference between correlated proportions or percentages. Psychometrika, 12(2): 153–157.

---

### Official Review · Reviewer_bzCu · 2023-08-11

**Soundness:** 3

**Excitement:**

2: Mediocre: This paper makes marginal contributions (vs non-contemporaneous work), so I would rather not see it in the conference.

**Paper Topic And Main Contributions:**

This paper introduces thee concept of transformer based text embedding depth also called TTE depth which is a way to rank embeddings based on angular distance within a corpus. They introduce a use case of few shot examples in prompt creation and use the metric to pick the samples. They show an improvement in F1 in 4 out of 6  classification they picked. They also use the metric to measure distance between synthetic and human generated data .

**Questions For The Authors:**

1.Have you tried this measure on other tasks that could use that relies more heavily on corpus distribution other than classification?

2.Prompt engineering is a significantly important task and a lot of papers have concluded that human curated prompts, highly tailored to each task are better than automated ones. Do you have plans to compare your automated prompts compared with human curated ones?

**Reasons To Accept:**

They are thorough in performing an array of ablation experiments to determine the optimal number of samples needed to determine the effectiveness of the metric .

The metric is a good way to measure distributional shift between human and synthetic data.



**Reasons To Reject:**

Their improvements in F1 on avg is not statistically significant enough to consider it better than just random.


**Reproducibility:**

3: Could reproduce the results with some difficulty. The settings of parameters are underspecified or subjectively determined; the training/evaluation data are not widely available.

**Reviewer Confidence:**

3: Pretty sure, but there's a chance I missed something. Although I have a good feel for this area in general, I did not carefully check the paper's details, e.g., the math, experimental design, or novelty.

---

> ### Author Rebuttal · Authors · 2023-08-28
>
> # Summary
> We thank the reviewer for their thoughtful comments. It was encouraging to see that the reviewer finds our ablation experiments thorough and interesting, in addition to recognizing the usefulness of TTE depth as a method for measuring distributional shift between corpora. We appreciate the reviewer’s feedback on our evaluation of TTE depth for the ICL prompt creation task. We answer specific questions and feedback from the reviewer below.
>
> # Questions
> > **Q1:** Their improvements in F1 on avg is not statistically significant enough to consider it better than just random.
>
> **A1:** We appreciate the reviewer’s feedback on the evaluation of the ICL prompt creation use case of TTE depth. To address this feedback, during the rebuttal period we have obtained new results using the full evaluation datasets, as opposed to samples of each evaluation dataset, for each of the 6 text classification tasks described in Section 4.2 of our paper. Additionally, we have statistically analyzed these new results using McNemar’s test [1]. As a reminder for the reviewer, we compare 2 TTE depth-based ranking strategies, depth-first (DEEP) and depth-first with label distribution matching (DLDM), against 2 baselines, random (RAND) and label distribution matching (LDM). We report average F1 scores on ICL classification using 5, 8, and 11 samples selected using each ranking strategy. For each of the depth-based strategies, we report whether the results were statistically significantly better than each of the baseline strategies (p < 0.05) using the signifier (Y/N) for whether the results were significantly better than the RAND and LDM baselines, respectively. Results are presented in the table below.
>
> |  Task |   RAND   |     LDM    |             DEEP            |             DLDM            |
> |:-----:|:--------:|:----------:|:---------------------------:|:---------------------------:|
> |       |    F1    |     F1     | F1 (Sig vs RAND/Sig vs LDM) | F1 (Sig vs RAND/Sig vs LDM) |
> | COLA  |  0.8164  |   0.8094   |         0.8166 (N/N)        |     **0.8220** (N/**Y**)    |
> | MRPC  |  0.7166  |   0.7250   |       **0.7334** (N/N)      |         0.6841 (N/N)        |
> | SST2  |  0.9486  |   0.9514   |       **0.9518** (N/N)      |         0.9414 (N/N)        |
> | RTE   |  0.6366  | **0.7352** |         0.4413 (N/N)        |         0.7258 (N/N)        |
> | THATE |  0.6039  |   0.6629   |   **0.7072** (**Y**/**Y**)  |     0.7009 (**Y**/**Y**)    |
> | TOFF  |  0.7361  |   0.7189   |   **0.7461** (**Y**/**Y**)  |     0.7411 (**Y**/**Y**)    |
> | _AVG_ | _0.7431_ |  _0.7671_  |           _0.7332_          |         **_0.7692_**        |
>
> We highlight that TTE depth-based ranking strategies select better samples for downstream ICL classification performance on 5 of the 6 full evaluation datasets. Additionally, the DLDM ranking strategy outperforms both baselines on average. Finally, in 9 of the 24 pairwise comparisons (including all 8 across the 2 social media tasks) the performance of depth-based ranking strategies is statistically significantly better than baseline approaches. We hope that these newly-obtained results using full evaluation datasets, in addition to the statistical significance testing, demonstrate the usefulness of TTE depth in the ICL prompt creation task. We will add these new results to Section 5 of the paper.
>
> We believe that this evaluation, in addition to the theoretical introduction to TTE depth and the evaluation of TTE depth as an inference method described in our paper, are sound contributions to corpus-level distributional inference and modeling in NLP.
>
> > **Q2:** Have you tried this measure on other tasks that could use that relies more heavily on corpus distribution other than classification?
>
> **A2:** Thank you for the great suggestion! The goal of this work is to introduce the use of TTE depth for modern NLP pipelines, hence we decided to thoroughly explore ICL prompt creation for text classification tasks as one use case, using multiple datasets from various domains. As an additional use case, we evaluate on the important inference task of measuring distributional shift between human and synthetic texts. However, in the future we plan to thoroughly explore other tasks. In particular, we agree with the reviewer and we hypothesize that tasks which are more corpus-dependent and use document-ordering strategies would be appropriate for future research, e.g., anomaly detection, or ICL prompt creation for information extraction. We hope that the release of the TTE depth python package, alongside this theoretical introductory work, will facilitate such interesting research in the future.
>
> > **Q3:** Prompt engineering is a significantly important task and a lot of papers have concluded that human curated prompts, highly tailored to each task are better than automated ones. Do you have plans to compare your automated prompts compared with human curated ones?
>
> **A3:** In short, yes. That’s a great idea and we will explore that in future explorations of TTE depth. We also want to share a bit more about the prompt creation process to address potential confusion.
>
> Following the terminology “prompt generation” introduced in prior work [2] we use the task definition of “prompt creation” as “the task of selecting representative examples from a data pool to prompt a LLM for labeling an unseen test example.” This is why we compare only against computational methods that automatically select representative texts from a larger training corpus, as in more recent works [3] and [4].
>
> While we focus on the task of selecting representative examples from a data pool to include in an ICL prompt, as each ICL prompt includes a human-written instruction in addition to selected examples, there is still room in our approach to curate human input in the prompt instruction. As our approach does not automatically write this instruction, we are not claiming a fully automated prompt selection pipeline. We think it would be an interesting experiment to compare our approach with other human-curated approaches where humans select representative examples.
>
> We also wish to emphasize that our goal in this work is not to achieve SOTA on the prompt creation task, but rather to theoretically introduce and thoroughly investigate the use of TTE depth as a tool for modern NLP modeling and inference tasks.
>
> # Conclusion/References
> Thank you for your thoughtful review and comments. We hope this response satisfies your concerns and resolves any potential ambiguity, please let us know if you have any other questions!
>
> [1] McNemar, Q. 1947. Note on the sampling error of the difference between correlated proportions or percentages. Psychometrika, 12(2): 153–157.
>
> [2] Tianyu Gao, Adam Fisch, and Danqi Chen. 2020. Making pre-trained language models better few-shot learners. arXiv preprint arXiv:2012.15723.
>
> [3] Zhou, Yuhang, Suraj Maharjan, and Beiye Liu. "Scalable Prompt Generation for Semi-supervised Learning with Language Models." Findings of the Association for Computational Linguistics: EACL 2023. 2023.
>
> [4] Shrivastava, Disha, Hugo Larochelle, and Daniel Tarlow. "Repository-level prompt generation for large language models of code." International Conference on Machine Learning. PMLR, 2023.

---

### Meta-Review · Area_Chair_5rCy · 2023-09-13

**Recommendation:** 3

**Metareview:**

The authors propose transformer-based text embedding (TTE) depth, and illustrate the use of this statistical depth on NLP tasks including in-context learning prompt creation and difference measurement between human-written and machine-generated text. The performance of TTE depth shows small improvement over baselines.

While the reviewers were happy with the methdology, the impression from the evaluation results were mixed. The response was able to allviate only some of the concerns.

---

### Decision · Program_Chairs · 2023-10-07

**Decision:**

Accept-Main

**Comment:**

The authors propose transformer-based text embedding (TTE) depth, and illustrate the use of this statistical depth on NLP tasks including in-context learning prompt creation and difference measurement between human-written and machine-generated text. The performance of TTE depth shows small improvement over baselines.

While the reviewers were happy with the methdology, the impression from the evaluation results were mixed. The response was able to allviate only some of the concerns.